

# Natural grazing by horses and cattle promotes bird diversity in a restored European alluvial grassland

Lilla Lovász[1,2], Fränzi Korner-Nievergelt[2,3] and Valentin Amrhein[1,2]

[1] Research Station Petite Camargue Alsacienne, Saint Louis, France
[2] Department of Environmental Sciences, Universität Basel, Basel, Switzerland
[3] Oikostat GmbH, Ettiswil, Switzerland

## ABSTRACT

**Context.** A challenge in grassland conservation is to maintain both the openness and the heterogeneity of the habitat to support the diversity of their animal communities, including birds—a taxon that is known to be sensitive to disturbance. An increasingly used management tool in European grassland conservation, especially in rewilding projects, is grazing by large herbivores such as horses and cattle. These grazers are believed to create and maintain patchy landscapes that promote diversity and richness of other species, but their influence on birds is often debated by conservationists, who raise concerns about the impact of disturbance by the grazers.

**Objectives.** Our aim was to examine the relationship between the abundance and species richness of birds across four foraging guilds and the area utilization patterns of Highland cattle and Konik horses in an alluvial grassland in France. We also aimed to examine the influence of land cover and season on the spatial distribution, including abundance and species richness, of different bird guilds present in the grazed area.

**Methods.** We used GPS-collars on all grazers and recorded their positions on an hourly basis over a study period of 1.5 years, assessing patterns of area usage. We counted birds weekly along three transects to describe their distribution within the grazed area and carried out land-cover surveys to describe the habitat. To assess how species richness and abundance of birds of different guilds were related to grazer density, season, and habitat characteristics, we used GAMM models in a spatially explicit framework. We also compared bird numbers at our main study site with a nearby non-grazed control area.

**Results.** The number of birds in the grazed area was about twice the number in the non-grazed control area. Within the grazed area, the abundance of open-area foraging birds increased with increasing grazer density. The number of woodland-foraging birds was also positively correlated with grazer density but less so than open-area foraging birds. The number of individuals in the aerial and wetland bird guilds was not correlated with the density of grazers. Most bird species and individuals were observed on open landscapes scattered with woody patches and waterbodies, and on areas with moderate grazer density.

**Conclusions.** Low-intensity grazing represents a potentially important management tool in creating heterogeneity in alluvial grasslands, thereby promoting suitable habitat for a diverse assemblage of bird species.

Corresponding author
Lilla Lovász, lovasz.lilla@yahoo.com

# INTRODUCTION

Year-round grazing by free-roaming large herbivores is increasingly used in the management of European grassland restoration projects (*Gilhaus, Stelzner & Hölzel, 2014*), yet conservationists often question whether it is an approach that benefits the maximum number of levels of the trophic chain. In the lowlands of Europe, grassland ecosystems were historically (*i.e.,* preceding the start of agriculture around 5500 BC) often shaped by river dynamics and activities of animals such as beavers and large herbivores (*Hejcman et al., 2013*). Until the recent increase in anthropogenic land use, such alluvial areas were ecosystems of high conservation value, hosting diverse assemblages of species, including many birds with various habitat and foraging needs (*Emanuelsson, 2008*; *Silva, 2008*).

However, ongoing human impact during the Anthropocene (*Lewis & Maslin, 2015*), such as river regulations and agricultural intensification, persist in driving the decline of alluvial grasslands and their associated species (*Bissels et al., 2004*; *Prach, 2007*). For example, a large-scale 10-year study found that more than 70% of grassland- and farmland-associated bird species were declining in number in Europe (*Donald et al., 2006*). Therefore, the European LIFE program focuses on the restoration of alluvial grasslands (*Silva, 2008*). The integration of grazing by ungulates, including horses and cattle emerges as an important management tool, particularly in light of the expanding scope of rewilding initiatives (*Linnartz & Meissner, 2014*; *Naundrup & Svenning, 2015*; *Pereira & Navarro, 2015*).

The diverse space-use patterns and grazing behaviour of ungulates create and maintain heterogenous landscapes (*Adler, Raff & Lauenroth, 2001*; *Fuhlendorf et al., 2006*; *Svenning, 2002*; *Vera, 2000*). In this context, heterogeneity refers to the structural complexity of the land cover, leading to richness in resources (reviewed by *Stein, Gerstner & Kreft, 2014*). Since more resources can support more individuals and species (*Hurlbert, 2004*; *Srivastava & Lawton, 1998*), grazed landscapes likely host a variety of bird species (*May, 1986*; *van Klink et al., 2016*; *VanWieren, 1995*). However, it has been difficult to differentiate between the effect of landscape structure (*Cody, 1985*) and the effect of grazers on the species richness and abundance of birds in grazed areas (*Hildén, 1965*; *Lovász, Korner-Nievergelt & Amrhein, 2021*; *Skórka, Martyka & Wójcik, 2006*).

Another layer of complexity comes from variation in the responses of birds to ungulate grazing, as these responses are shaped by the intensity and seasonal variation of grazing, and by the foraging and breeding ecology of the bird species. Grasslands grazed at low intensity (<0.5 animal units per hectare, where 1 animal unit = 500 kg; *Andresen et al., 1990*) seem to benefit many grassland-associated bird species, enhancing foraging and nesting opportunities (*Buckingham & Peach, 2005*; *Pärt & Söderström, 1999*). For instance, the abundances of the Eurasian Skylark *Alauda arvensis*, Red-backed Shrike *Lanius collurio* or Meadow Pipit *Anthus pratensis* were found to be higher with low-intensity grazing in the breeding season (*Ceresa et al., 2012*; *Evans et al., 2006*; *Kohler, Hiller & Tischew, 2016*). This

is likely because birds profit from increased food availability resulting from the grazers' activity, yet the evidence for such pattern is indirect. For instance, *Dennis et al. (2008)* found that structural diversity of the vegetation increased foliar arthropod abundance, and there is also evidence that food availability (such as insects attracted by grazers) is a crucial factor influencing bird abundance (*Vickery et al., 2001*; *Atkinson et al., 2005*). Furthermore, *Evans et al. (2006)* found that low-intensity grazing by mixed-species livestock (cattle and sheep) improved the abundance of an insectivorous bird, the Meadow Pipit. Similarly, a study on the impact of single-species grazing by feral horses in Argentina found that areas with moderate horse densities had the highest species richness and abundance of birds (*Zalba & Cozzani, 2004*). On the other hand, intense grazing (>2.0 animal units per hectare) likely only favours bird species that exploit very short vegetation to feed on soil invertebrates, such as Thrushes, *Turdus sp* (review by *Buckingham & Peach, 2005*). In general, species richness among birds associated with open areas seems to be higher in areas grazed with low intensity, and lower in areas with high grazer densities (*Barzan, Bellis & Dardanelli, 2021*; *Batary, Baldi & Erdos, 2007*).

The evidence for a relationship between grazers and bird species that are not heavily reliant on open areas remains ambiguous. While wetland-associated birds might face challenges due to the trampling of nests around water bodies where ungulates accumulate for drinking (*Mandema et al., 2013*; *Pakanen, Luukkonen & Koivula, 2011*), the heterogenous vegetation maintained by grazers can also facilitate nesting, as it minimizes predation risk (*Whittingham & Evans, 2004*).

Studies on seasonal variation in the influence of grazers on birds often found contradictory results, describing both positive and negative relationships between birds and grazers, even within the same species (reviewed by *Atkinson, Buckingham & Morris, 2004*). For instance, *Wilson, Taylor & Muirhead (1996)* observed Skylarks avoiding grazed fields in winter and reported "extremely low densities" of Skylarks on grazed pastures in summer as well (*Wilson et al., 1997*). In contrast, *Suárez, Garza & Morales (2003)* emphasized the importance of grazed areas for both wintering and breeding Skylarks, as they found a positive association between Skylark presence and pastures with short vegetation in both seasons.

Previous studies primarily compared the impacts of grazing on birds between enclosures with different grazing intensities, (*e.g.*, *Baldi, Batary & Erdos, 2005*; *Söderström, Pärt & Linnarsson, 2001*) and often only in experimental conditions (*Evans et al., 2006*). However, the grazers' uneven distribution within enclosures due to their heterogeneous use of an area complicates quantifying actual grazing pressure (*Fern et al., 2020*). Thus, the responses of birds to such varying area usage by grazers have rarely been addressed (*Lovász, Korner-Nievergelt & Amrhein, 2021*), and real-life rather than experimental conditions have been understudied in the past. Furthermore, previous research mainly focused on the effects of grazing on breeding birds (*e.g.*, *Coppedge et al., 2008*; *Kohler, Hiller & Tischew, 2016*; *Vold, Berkeley & McNew, 2019*) or on specific bird guilds (*e.g.*: *Cox et al., 2014*; *Lituma et al., 2022*), rather than studying the dynamics of larger bird communities.

In this study, we aimed to address this gap by investigating how the species richness and abundance of individuals of a bird community is related to the spatial distribution

of a mixed assemblage of Highland cattle and Konik horses in a natural grazing regime throughout the year. We focussed on the relationships between the numbers of bird individuals, incorporating species within four guilds, and the density of cattle and horse GPS positions within our study area, which is a recently restored alluvial grassland in a French nature reserve. Additionally, we evaluated the dynamic interplay between bird and grazer densities across seasons and varying land-cover conditions. We also compared the species richness and abundance of birds in the grazed area to a non-grazed control site in the breeding season. We expected that the presence of birds within the grazed area would show variation depending on the changes in grazer densities, habitat characteristics and season, and according to the particular ecology of birds in different foraging guilds. We also expected that the grazed area would attract more bird species and individuals than the non-grazed area and that grazing exerts the most important influence on the density of birds foraging in open areas. (Portions of this text were previously published as part of a thesis (*Lovász, 2022*) and a preprint: *Lovász, Korner-Nievergelt & Amrhein, 2024*).

## MATERIALS AND METHODS

### Study site

We carried out the study in a 32-ha test-enclosure that is part of an ecosystem restoration project of the nature reserve Petite Camargue Alsacienne, located in Alsace, France, north of the city of Basel, Switzerland. The area is situated on the Rhine Island (see the map of the study area and further details in *Lovász, Korner-Nievergelt & Amrhein (2021)*), where approximately 100 ha of former agricultural fields had been transformed into an alluvial environment. The ecosystem restoration process started in 2014 and involved establishing a heterogenous grassland scattered with shrubs (hawthorn, *Crataegus sp*; dog rose, *Rosa canina*), young and old trees (willow, *Salix sp*; oak, *Quercus sp*) and bare-ground gravel sites, surrounded by patches of old forests (oak; ash, *Fraxinus sp*). Water from the Rhine river is diverted through the area as a small stream, and ground-water ponds have also been created. The aim has been to maintain the heterogeneity of the area and to promote a species-rich alluvial ecosystem.

The study area has been managed using the approach of natural grazing, a low-intensity year-round mixed grazing regime (<0.5 animals per ha), with the aim of substituting extinct wild herbivores, such as the wild horse *Equus ferus* or the aurochs *Bos primigenius*, with domestic breeds kept in semi-wild conditions (*Lovász, Korner-Nievergelt & Amrhein, 2021*). This natural grazing approach includes minimal human intervention and no systematic winter feeding (*Linnell, Kaczensky & Lescureux, 2016*; *Vermeulen, 2015*).

### Land-cover characteristics

We carried out surveys to characterize the landscape on the grazed study site and the non-grazed control area. We used satellite imagery (Google Maps) in QGIS (version 3.4.4-Madeira) to divide the study area into $50 \times 50$ m grid cells. With the help of a printed map of these grid cells (using the QGIS printout layer) and a handheld GPS (application: Gaia GPS), the observer (L.L.) visited each grid cell. The observer visually estimated the percentage of types of land cover inside the cell by marking the corner points with flags
and standing in the midpoint of the cell (which was already indicated with permanent markings used in a different study). When it was necessary for adequate estimation of the cover types (for example, when visibility decreased due to trees), the observer walked along one diagonal of the grid cell to determine the cover attributes. Additionally, the observer compared the observations to the QGIS map image of the area while estimating the percentages. The comprehensive land-cover survey took place once during the study period in summer 2020. The observed large-scale habitat characteristics did not noticeably change during the 1.5-year study period, as indicated by weekly visual inspections.

We followed *Cunningham & Johnson (2019)* when defining a small number of categories for land-cover types in our study area: tree-cover (all trees of ca. ≥ 3 years old), sapling-cover (all growth stages of young trees of ca. <3 years old), shrub-cover, cover of meadow, surface of bare ground, and surface of water on the respective grid cell. Based on *Hildén (1965)*, these characteristics were suggested as important determinants for habitat selection of birds in our study site.

We identified six different land-cover types, indicating a heterogenous mosaic-like environment. The cover by meadow (grassland) was the most abundant (61%). Bare ground, tree cover and shrub cover represented only a small proportion of all cover types (6.9%, 3.4%, and 1.7%, respectively), as the area was restored with the aim of recreating an open alluvial grassland. Single trees and shrubs were patchily distributed without providing extensive cover. The relatively high percentage of sapling cover (17.9%) consisted mainly of poplar (*Populus sp*) trees. Such sapling patches were often relatively large (up to more than 1 ha), with either bare ground or species-poor meadow patches as understory. Extensive areas with bare ground were only found near water bodies. Water bodies represented 9.2% of all cover types, and all of them were groundwater ponds with relatively stable water levels throughout the year.

## Cattle and horse data

To counteract the natural succession by willow and poplar saplings encroaching upon the meadows and to foster a heterogenous environment, large grazers were progressively introduced into the test enclosure. The introduction began with horses in September 2018, followed by cattle in January 2019. The initial number of five Highland cattle and five Konik horses increased to seven cattle (five adults and additionally two calves born on-site) in the summer of 2019, and to seven horses (five adults and additionally two young mares brought from other areas) in the summer of 2020. Over the duration of our data collection, the overall stocking rate increased from ∼0.3 animals per ha to ∼0.4 animals per ha.

To determine the density distribution of the horses and cattle across the study area, we recorded hourly GPS positions of the grazers and considered these as a proxy for the space use of the animals. All grazers (14 in total) were equipped with GPS collars (Followit, type Pellego) upon their arrival to the study area by the management of the nature reserve. The collars registered the position of the animals once per hour. The first round of data collection was from January to July 2019 (the GPS data from this first data collection were used in *Lovász, Korner-Nievergelt & Amrhein, 2021*). The GPS collars recorded for 5 to 6 months with one charge, and it was only possible to subsequently recharge the devices of the

horses; the cattle were not tame enough to handle the collars without the need of capturing and blocking the animal (which was only possible in winter during the regular roundup of cattle). Thus, more durable GPS collars (Followit, type Tellus) were used for two cattle starting from March 2020, instead of the five formerly used collars (the structure of the collected data of the two collars was identical). Data were then continuously collected until March 2021. In total, data collection spanned one year and six months. In our analysis, we accounted for different numbers of collars during the different time periods.

The data were downloaded through satellite processing from the interface of the GPS collar provider (Followit, Lindesberg, Sweden), and, thus, no contact with the animals was necessary to access the data. GPS collars comply with animal welfare requirements for horses (*Collins et al., 2014*; *Hennig, Beck & Scasta, 2018*) and have been widely used also on cattle for decades, without causing harm or disturbance (*Turner et al., 2000*; *Ungar et al., 2005*).

The GPS positions in our dataset may show some imprecision because GPS accuracy can vary depending on atmospheric conditions or due to satellite or receiver errors (*Hurn, 1993*), caused by satellite geometry (*Dussault et al., 2001*), topography, overhead canopies, or adjacent structures (*Di Orio, Callas & Schaefer, 2003*; *Moen et al., 1996*). The GPS collars did not record HDOP (horizontal dilution of precision), so we could not statistically account for inaccuracy (*Langley, 1999*). However, as the rate of GPS positions falling outside of the fenced area were negligible (see *Lovász, Korner-Nievergelt & Amrhein, 2021*), we assumed that imprecision would not strongly influence our results.

## Bird abundance and species richness data
### Main study site
We surveyed bird abundance by transect counting following the recommendations of *Gregory, Gibbons & Donald (2004)* and *Bibby et al. (2000)*. Our methods were similar to those in *Lovász, Korner-Nievergelt & Amrhein, 2021*, but to facilitate understanding, we give detailed descriptions here as well.

Our objective was to conduct surveys one morning every two weeks from 31 January 2019 to 24 July 2019, and weekly between 20 March 2020 and 22 March 2021. Data collection took place exclusively under favourable weather conditions (absence of precipitation, wind speed below 5 m/s). Our total sampled area was 21.4 ha. We did not include the 10.6-ha forested area of the 32-ha enclosure, as our aim was to investigate bird abundance and species richness in the open areas.

We established three line transects over the grazed open area of the test enclosure, each approximately 700 m long, in a way that provided visual and/or auditory coverage of all parts of the area (map of the study area: Fig. S1). A trained observer (L.L.) walked along each transect at a slow pace and recorded the position of all identified individual birds (seen or heard) on a digital map (Map Marker 2.11_1442). Birds flying higher than 20 m above the ground were not recorded except if they showed some connection to the area (*e.g.*, Skylarks *Alauda arvensis* that made territorial song-flights at $\geq 20$ m elevation were counted, but raptors or water birds crossing $\geq 20$ m over the meadows or migrating Common Swifts *Apus apus* were not).

During each survey, all three transects were visited between sunrise and noon, excluding dawn to minimize differences in detectability due to rapid changes in the birds' conspicuousness and activity (*Dawson, 1981*). We systematically varied the order of surveying the three transects among the surveys. We assumed that differences in bird detectability between the three transects were minimal, due to the similar open habitat of the surveyed areas.

We avoided visual and auditory double counts as much as possible by using a detection distance of about 60 m towards each side of the transect, which also corresponded to the assumed range of visual detectability of birds on the ground. By counting birds only inside these 120 m-wide stretches, the surveyed areas did not overlap, but still covered the entire study site. We followed the recommendation of *Dawson & Bull (1975)* by considering observations as different individuals only if the observer was reasonably certain that the same individual was not observed.

We classified the observed bird species into four different guilds (functional groups on an ecological basis *Wilson, 1999*) according to their foraging ecology (*Francis et al., 2020*): 'aerial', 'open-area', 'wetland', 'woodland' guilds (see Table S1). For the classification, we considered the primary foraging habitats on the Rhine Island and used the concept of a 'guild' as described by *Root (1967)*, which refers to a "group of species that exploit the same class of environmental resources in a similar way".

### Control area

We also carried out transect surveys on a non-grazed control area that consisted of two parts, one 14.3 ha and the other 9.5 ha (in total 23.8 ha), which we selected on two different regions of the Rhine Island, each neighboring the main study site, and each having similarly mainly open habitats. We considered the two parts together as one control site and defined three transects for transect sampling, each of approximately 700 m length, similar to the main study site. The control site included similar habitats as our main study area but was not grazed by cattle or horses during our data collection period. Notably, the control area was grazed by sheep and goats during a few weeks in spring and summer 2019, but not in 2020. Therefore, we considered this area as a non-grazed control site in 2020. Our survey method was analogous to that of the main study site. Data collection took place between 8 April 2020 and 22 May 2020 (as the site was available for research purposes only during these two months due to an exceptional permission resulting from the COVID-19 lockdown).

## Permissions

The fieldwork was carried out with the permission of the national nature reserve Petite Camargue Alsacienne.

## Statistical analysis

For the analysis, we used the $50 \times 50$ m grid cells of the Universal Transverse Mercator (UTM) coordinate system's grid over the study area. We first calculated the mean numbers of species and of individuals in the main study site and the control site during the breeding season. Subsequently, to assess how bird abundance and species richness were related to

grazer density, we used only data from our main study site (since the control site was not grazed).

To analyze how the abundance of birds of different guilds were related to grazer density, season, and habitat characteristics, we used generalized additive mixed models (GAMM) that we fitted to the data of each bird guild separately. Our response variable was the number of bird individuals counted per survey per $50 \times 50$ m grid cell, summed over all species belonging to a bird guild. First, we fitted a negative binomial model, but a posterior predictive check showed that the data contained too many zeros. We therefore assumed a zero-inflated negative binomial distribution of the bird counts with a constant proportion of zero values. We used the logarithm as link function for the count model. As predictors, we used grazer density, date (day of year), and habitat characteristics.

Grazer density was measured as the sum of the hourly horse and cattle GPS positions per grid cell over the 3 weeks prior to a bird survey. Our previous study (*Lovász, Korner-Nievergelt & Amrhein, 2021*) showed that the effects of horse *vs* cattle densities on bird count densities were rather similar, and, therefore, we pooled the data for horse and cattle GPS positions. We further assumed that grazer space-use patterns earlier than 3 weeks before the respective bird survey did not substantially influence space use of birds (based on *Lovász, Korner-Nievergelt & Amrhein, 2021*; this was an arbitrary cut-off time we chose for pooling grazer positions). Because we assumed that the detectability of birds was relatively homogeneous across the main site and control site areas, and our aim was not to estimate the total number of birds present, we did not take detectability into account in our analyses, following *Buckland et al. (2001)*. As habitat characteristics, we used the land-cover variables "bare ground", "sapling", "shrub" and "tree". To avoid redundancy in the predictor variables due to cover types summing up to 100%, we did not include the most abundant land cover type "meadow". The cover type "water" was included as a binary variable indicating presence or absence of water bodies on a 50x50 m cell. All predictors were z-transformed so that their means were zero and their standard deviations were one.

We used a two-dimensional tensor product smooth for grazer density and day of year. To obtain the smooth terms, we used three cubic regression splines, as implemented in the function t2 of the package mgcv (*Wood, Scheipl & Faraway, 2013*). Because some grid cells at the edges of the study area had sizes smaller than $50 \times 50$ m, we used the logarithm of the grid cell size as an offset in the linear predictor of the counts.

The grid cell ID was first used as a random factor to account for repeated measures. However, Markov chains of these model fits did not converge well. We therefore omitted the random factor from the model. To assess the strength of the pseudoreplication that may have arisen by omitting the grid cell ID as a random factor, we estimated the proportion of residual variance that can be explained by among-grid cell variance using normal linear mixed models. This proportion was relatively low for all guilds (wetland birds: 8.1%, woodland birds: 2.6%, open-area birds: 0.7%, aerial birds: <0.1%). We thus considered that pseudoreplication due to repeated measures of grid cells was low in our study. However, we acknowledge in the interpretation of our results that our compatibility intervals (*Amrhein*

& Greenland, 2022) may underestimate statistical uncertainty, particularly for wetland birds.

To analyze how species richness was related to grazer density, day of year, and habitat characteristics, we used the number of species detected during one survey on each of the 50 × 50 m grid cells as the outcome variable in another GAMM. Similarly to the model for the abundance, here we also used a zero-inflated negative binomial model. We used the same predictors as those used for the analyses of abundance, except that the logarithm of the grid-cell size was used as a covariate instead of an offset. For analyzing species richness, we used the grid cell ID as a random factor, to account for repeated measures in the grid cells. Unlike for the abundance models, Markov chains for the species richness model including cell ID as random factor converged well.

All models were fitted using Hamiltonian Monte Carlo methods as implemented in Stan (mc-stan.org; *Carpenter et al., 2017*). We used the R-package brms as an interface to Stan (*Bürkner, 2017*). We used the default prior distributions that were optimized for z-transformed predictors and that did not markedly affect the results: the positive range of t (3, −2.3, 2.5) for the intercept of the negative-binomial model, a logistic (0,1) for the intercept of the zero-model, flat priors for all model coefficients, positive range of t (3,0,2.5) for the variance parameters (random effects and coefficients of the smoother), and gamma (0.01, 0.01) for the shape parameter of the negative-binomial model.

After fitting the models, we checked convergence graphically by looking at the Markov chains and using the R-hat value. We further inspected the goodness-of-fit *via* posterior predictive model checking using the package shinystan (*Gabry, 2017*). We compared density distributions of our data with density distributions of data replicated from the models, and we compared means, standard deviation, minimum and maximum of our data with replicated data based on the models. We also checked for spatial correlation by looking at the semi-variance and by visualizing the residuals on a map, and we did not detect conspicuous spatial autocorrelation.

Posterior predictive model checking revealed that a Poisson distribution or negative binomial distribution did not describe the data distribution well, but zero-inflated negative binomial distributions fitted well to all our response variables.

## RESULTS

### Bird counts on the main study site and the control site

In total, we made 5,447 observations of individuals of 87 bird species during the entire study (see Table S2 for the species list). On the main (grazed) study site, we surveyed 112 grid cells during a total of 73 surveys between 31 January 2019 and 22 March 2021. During these 73 surveys, we counted 5,104 individual birds.

For comparison, we conducted surveys at the (non-grazed) control site during a period of about two months in the breeding season, from the 89th to the 142nd day of the year. On this control site, we surveyed 121 grid cells during seven surveys and compared the number of individuals observed on the control site and the main site during the same period in 2019 (six surveys) and in 2020 (seven surveys). The mean (±SE) number of observed bird

individuals per survey per grid cell was 0.93 (±0.14) on the main site, and 0.43 (±0.08) on the control site. Also, the mean species richness (number of observed bird species per survey per grid cell) was higher in the grazed area (0.27 ±0.01) than on the control site (0.18 ± 0.01) during the period of comparison.

## Effects of grazer density, season, and habitat on bird abundance

In the following, we describe how the numbers of observations of aerial, open-area, wetland, and woodland bird individuals changed seasonally, in relation to grazer density (*i.e.,* the numbers of GPS positions of grazers per grid cell summed over a 3-week period) and in relation to the habitat characteristics for our main study site.

### Aerial-foraging birds

In total, we made 831 observations of aerial-foraging bird individuals. Almost all birds of this guild were swallows and swifts that do not breed at the study site (422 Barn swallows, 138 House martins, 269 Common swifts). During spring migration (April and early May), the numbers of individuals in the aerial foraging guild reached a peak, particularly in areas with low grazer density (Figs. 1, 2A and 2B). During the rest of the year, there was no obvious relationship between the numbers of aerial-foraging birds and grazer densities (Figs. 1, 2B; the apparent increase at high grazer densities on 1 September in Fig. 2A had high statistical uncertainty, as indicated by the wide compatibility interval). Aerial birds were observed in similar numbers (average 0.2 individuals per grid cell per survey: see the heatmap contour lines in Fig. 1) throughout the whole range of grazer density in the breeding season (April–July).

There were slightly more counts of aerial birds on plots with higher proportions of meadow cover (Fig. S2A). We expected to find more aerial-foraging birds over water bodies, since swallows often forage on insects above water. However, we did not find such a difference when comparing grid cells with and without water bodies. Also, the percentage of coverage in all other habitat categories showed almost no relationship with bird abundance in the aerial-foraging guild (Fig. S2A).

### Open-area foraging birds

In total, we made 2,352 observations of open-area foraging birds. We observed more individuals at grid cells with higher grazer frequencies throughout the year (Figs. 1, 2A–2B). The number of individuals reached its peak in spring and early summer (*i.e.,* during the breeding period; Figs. 1 and 2) and this peak was particularly strong with high grazer densities (Fig. 2B).

The most frequently observed birds in the open-area foraging guild were Starlings (940 individuals), Skylarks (639), and Pipits (473). Starlings were mostly non-breeding guests and were only present on the study site during spring and summer, while Skylarks were observed both breeding and migrating, and Pipits were migrants and occasional winter visitors. The peak of bird observations of this guild in spring (Fig. 2B) may, therefore, mainly be shaped by the spring arrival of Starlings.

When investigating the land-cover effects, the number of individuals in the open-area foraging guild slightly decreased with an increasing proportion of bare ground (Fig. S2B).

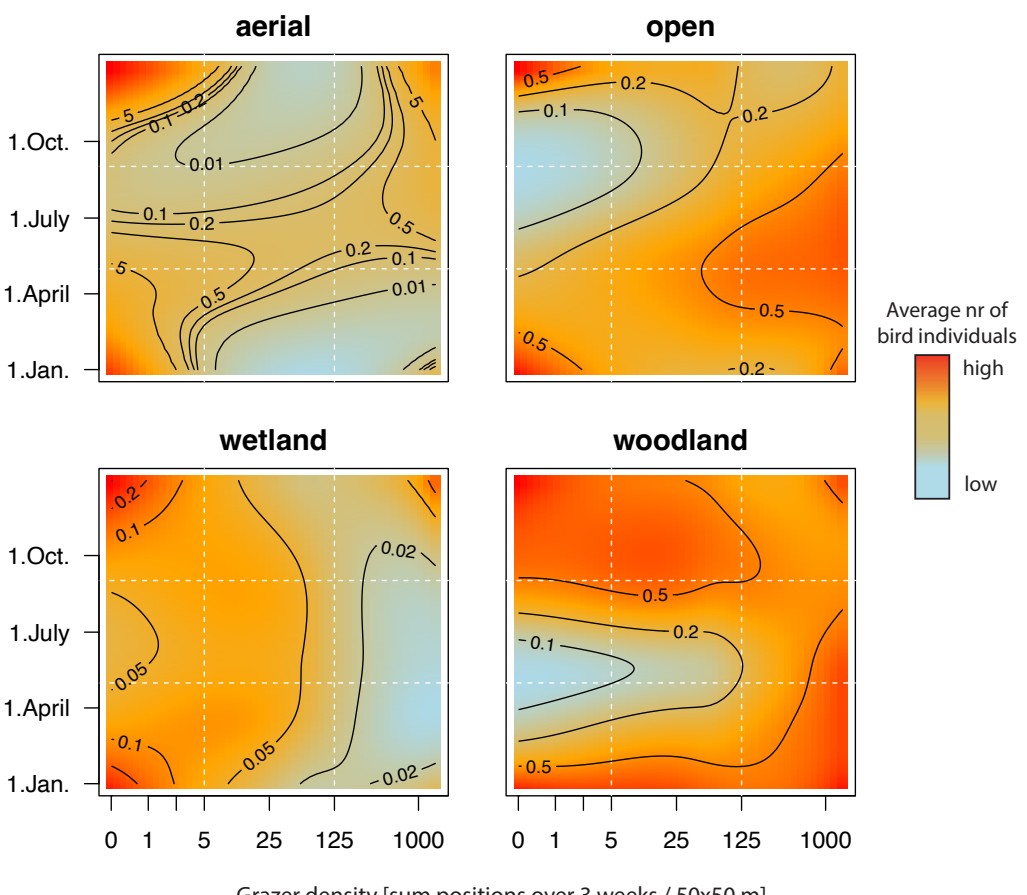

**Figure 1** **Number of individuals (abundance) of the four different foraging guilds of birds, in relation to grazer density and date.** Heatmap colors with contour lines indicate average number of individuals. Vertical dotted lines indicate segment lines of grazer densities of five and 125 GPS positions per grid cell per 3-weeks period, and horizontal dotted lines show 1 May and 1 September. These segments are plotted in Fig. 2, where we also provide 95% compatibility intervals. Note that patterns are unreliable at extreme values (in the corners of each plot).

This probably reflects the fact that in our study site, areas with bare ground were mainly found between and around sapling-covered areas; and sapling cover appeared to negatively affect the numbers of open-area birds (Fig. S2B). Statistical uncertainty (as indicated by the intervals) for the meadow, tree and shrub cover was too high to draw conclusions. There was only a negligible difference in the number of open-area foraging birds between grid cells with and without water bodies (Fig. S2B).

### Wetland-foraging birds

In total, we made 1,051 observations of wetland-foraging birds, from which the Mute swan (235 individuals), the Mallard (172) and the Tufted duck (140) were the most numerous, all of which bred at or around the study site in low numbers but were mostly non-breeding guests.
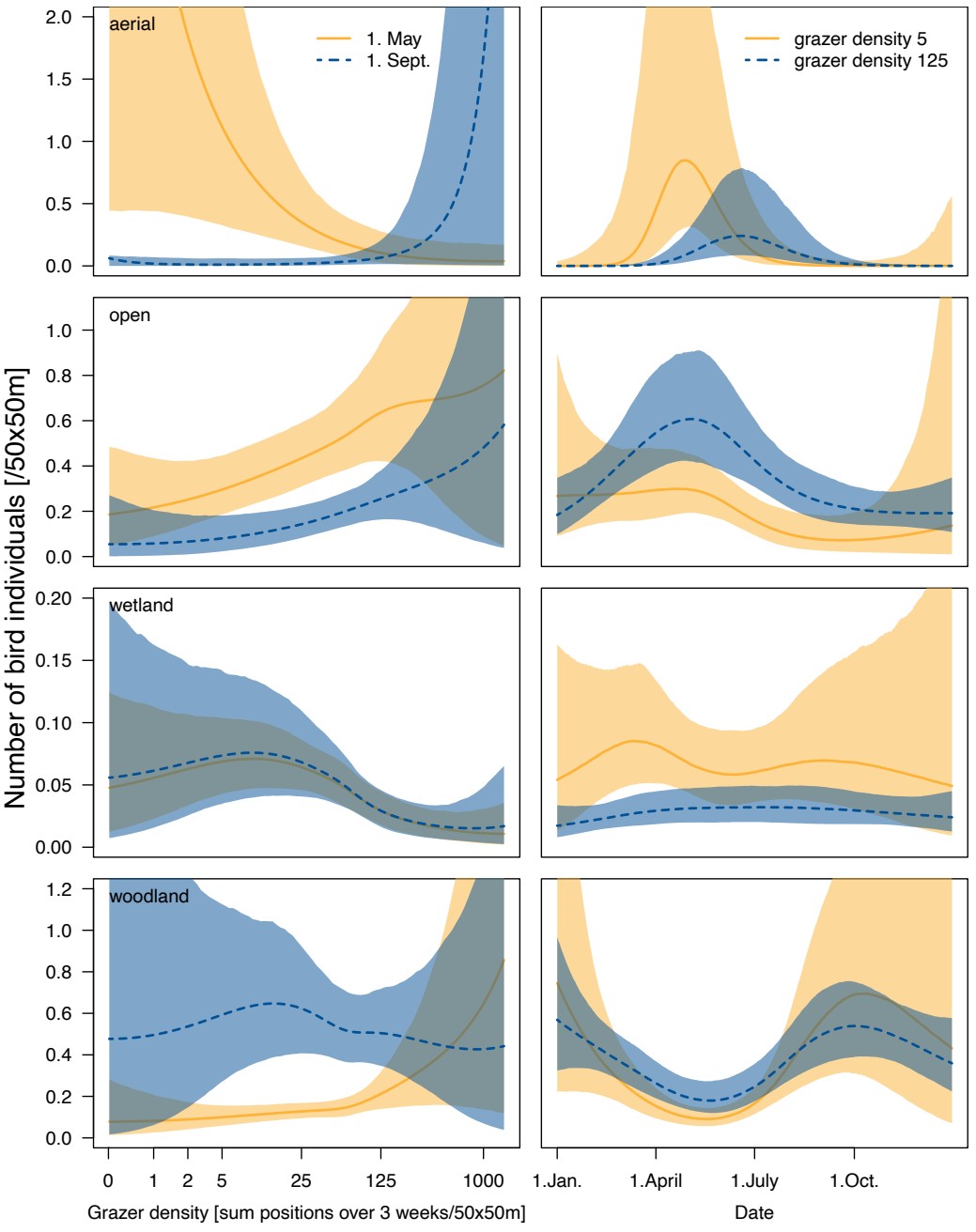

**Figure 2** **Number of individuals (abundance) of the four foraging guilds of birds in relation to grazer density and date.** (A) The segment of bird abundance for each guild for two haphazardly selected days: 1 May (breeding period) and 1 September (migration period), with 95% compatibility intervals (dotted lines). (B) The segment of bird abundance for each guild for two selected grazer densities: five GPS positions per grid cell per 3 weeks, meaning low density, and 125 positions per grid cell per 3 weeks, meaning high grazer density (average grazer density throughout the study period was 24.1 GPS positions per grid cell per 3 weeks). Dotted lines are 95% compatibility intervals. Note that patterns are unreliable at extreme values (at the beginning and end of the *x*-axis). (Colors of this figure were changed to color schemes corresponding to non-normal color vision using Adobe Illustrator version 28.4.1).

The number of individuals of wetland species was similar at low and moderate grazer densities and lower at high grazer densities throughout the year (Fig. 1).

When investigating which land cover type attracted wetland birds, we found that, as expected, birds of this guild were associated with water: we counted about ten times more individuals at grid cells with water bodies than without (Fig. S2C). The number of individuals slightly decreased with increasing bare ground and sapling cover, while it showed a moderate increase with increasing meadow, shrub, and tree cover (Fig. S2C). This pattern may reflect that in our study site, water bodies were often surrounded by meadows, and there were also some trees and shrubs that grew near these water bodies.

### *Woodland-foraging birds*

In total, we counted 1,213 woodland-foraging birds; the most numerous were the Great tit (281 individuals), the Common chaffinch (183) and the Red-backed shrike (140), all of which bred within or around the study site.

Overall, there were fewer woodland-foraging birds during the breeding period than during the migration period and during winter (Figs. 1 and 2). While during the breeding period, numbers of woodland birds were higher at higher grazer densities (Figs. 1 and 2A), during autumn and winter there was a minor increase in bird numbers at lower grazer densities (Fig. 1, but statistical uncertainty in Fig. 2A for September was high).

Woodland-associated birds used the different land cover types as expected: woodland birds were more often observed in areas with higher shrub and tree cover (Fig. S2D), while fewer individuals were observed in areas with a higher cover of bare ground, meadow, and saplings (Fig. S2D). We observed almost no difference between the number of woodland-associated birds in grid cells with or without water bodies (Fig. S2D).

## Effects of grazer density, season, and habitat on the species richness of birds

We found species richness to be relatively constant throughout the year, with an average of about 0.3 species counted per grid cell per survey (Fig. 3).

Independently of the land-cover variables included in the model, most species per grid cell were observed at average grazer densities (24.1 GPS positions per grid cell per 3 weeks) on the median day of the study period (1st June), and this pattern was similar for all seasons (Fig. 4 and see Fig. S3). However, the pattern at higher grazer densities was unclear due to the large compatibility intervals, at least for the median day of the study period (Fig. S3, right side).

The number of bird species decreased slightly with increasing cover of bare ground, meadow, and saplings (Figs. 5A/5C), while it increased with more extensive shrub and tree cover (Figs. 5D/5E) and was also higher in grid cells with water bodies (Fig. 5F).

## DISCUSSION

In this study, we investigated how mixed-species (horse and cattle) grazing at low to moderate intensities was related to the presence of birds in several feeding guilds. We aimed to describe the complex effects of the space use of grazers and the habitat on four bird guilds throughout the year in an alluvial ecosystem restoration site in France.
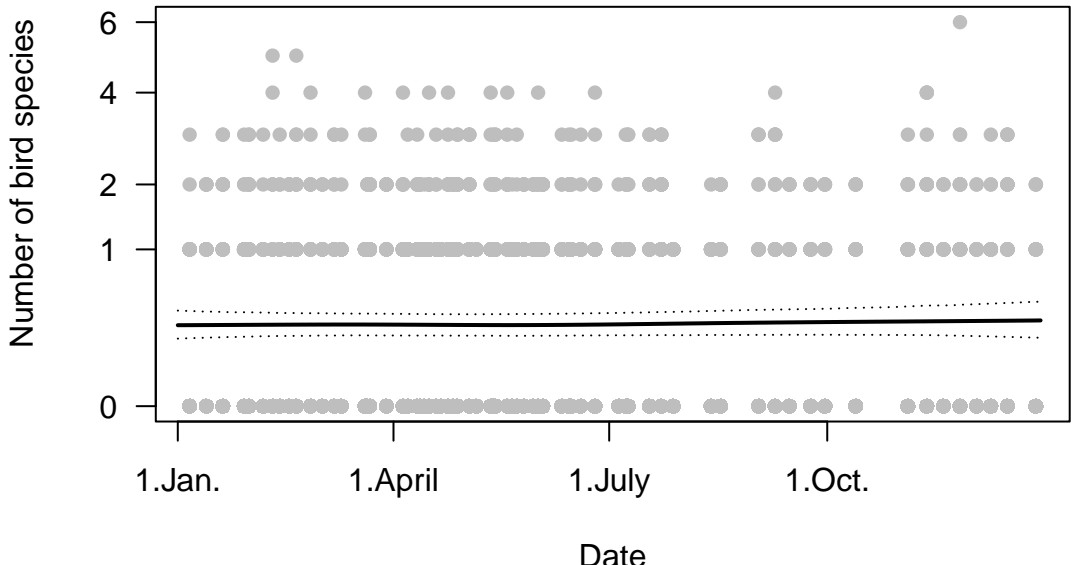

**Figure 3 Changes of number of bird species as a function of date.** Dotted lines are 95% compatibility intervals; grey dots indicate the number of species detected in a grid cell on a given day.

In line with our expectations, we found that the grazed area attracted, on average, more than twice as many individuals and significantly more species (close to double) than the non-grazed area. A similar pattern was observed in a riparian environment concerning livestock grazing in an experimental context (*Nelson, Gray & Evans, 2011*). However, to our understanding, our study marks the first attempt to compare areas with and without low-intensity mixed-species grazing in a nature conservation context. Our study also seems to be first in accounting for the spatio-temporal variation of grazer presence inside the grazed area instead of comparing only mean grazing intensities of areas with different numbers of grazers.

We found that the bird guild that showed the strongest positive relationship with low-intensity grazing by cattle and horses was the open-area foraging guild. Here, the number of observations was consistently higher at higher grazer density throughout the year. These patterns are likely due to the open habitat maintained by grazers (*Leal et al., 2019*), providing better food availability, prey visibility, and predator detectability for birds foraging in the open (*Buckingham & Peach, 2005*; *Martin & Possingham, 2005*; *Cox et al., 2014*; *Whittingham & Evans, 2004*). However, the density of birds associated with open areas in our study site was better explained by grazer density than by land cover (the relationship with grazers was similarly strong year-round, as heatmap colors suggest in Fig. 1, while the relationship with land-cover variables was less clear, as shown in Fig. S2B), suggesting that individuals of this guild closely responded to the actual presence of the grazers rather than to the land cover created by them. Furthermore, the open-area foraging birds were present in our study area year-round, yet in smaller numbers in the autumn and winter periods than in the breeding period. *Neilly & Schwarzkopf (2019)* similarly

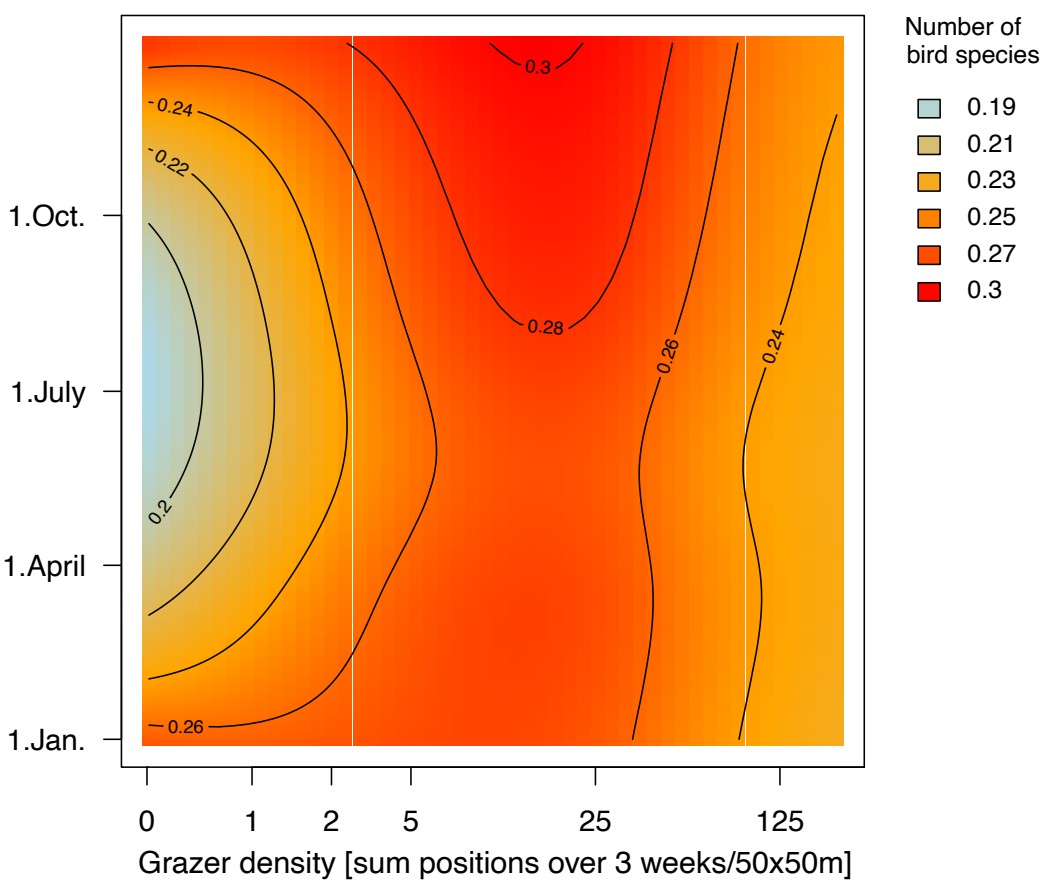

**Figure 4  Number of species in relation to grazer density and date.** Colors and contour lines indicate the average number of species per grid cell. (For the assessment of the compatibility intervals see Fig. S2).

pointed out that ground-foraging birds (which are also part of our open-area guild) are, among all guilds, the most responsive to grazers. This may be due to the generally higher insect abundance attracted by cattle and horses (*Dennis et al., 2008*; *Vickery et al., 2001*), or due to the insects flushed from the vegetation following the movement of herbivores (*Källander, 2004*). In particular, Starlings are known to associate closely with large herbivores (*Källander, 2004*), and this species was also the most abundant in the open-area guild in our study site. After Starlings, the most abundant species in the guild were Skylarks and Pipits, which are known to forage in areas with short and diverse vegetation maintained by grazers (*Buckingham, Peach & Fox, 2006*; *Chamberlain et al., 1999*; *Evans et al., 2006*; *Wakeham-Dawson & Aebischer, 1998*). The high abundance of these three species reflects the importance of areas with natural grazing management for the open-area foraging guild.

Woodland-foraging birds also showed a positive relationship with grazer space use, especially in winter, when they were present in the grazed area in higher numbers than in summer. The seasonal variation in the abundance of woodland birds and their relation to grazers may be explained by the fact that our study site is mainly an open area with few trees

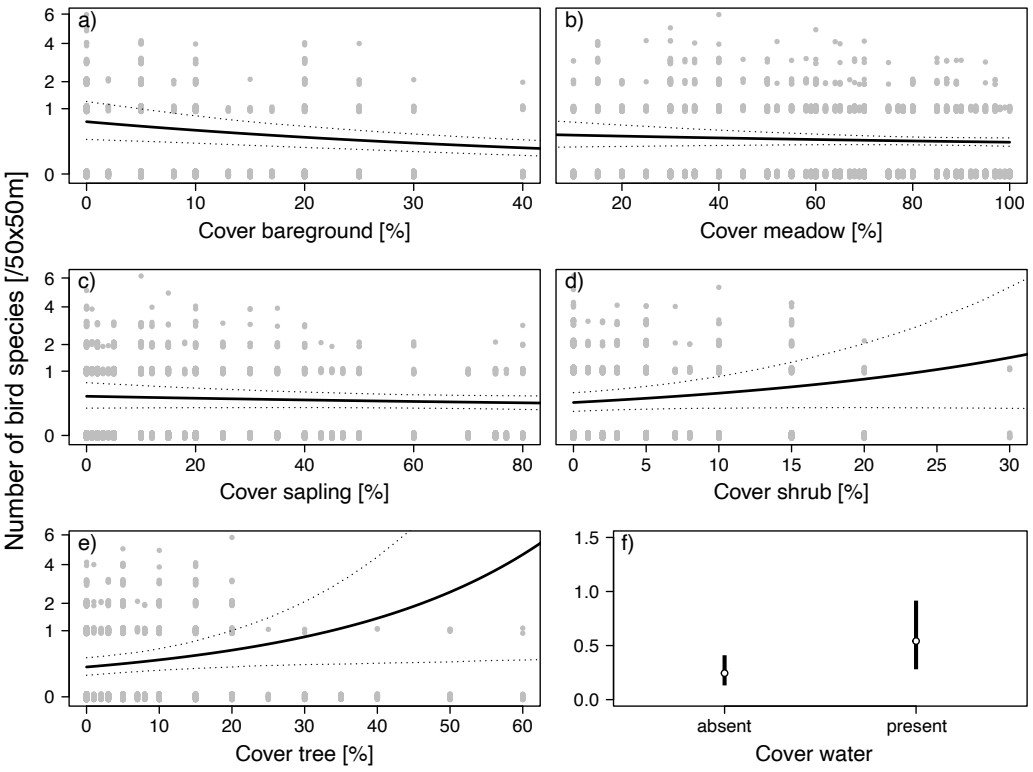

**Figure 5 Relationships between overall bird species richness and land cover types.** Note that the scale of the *x* axis on plots (A–E) changes according to the maximum percentage of the respective land cover type over all grid cells. Plot (F) depicts the binarized effect of water presence in the surveyed grid cells.

and shrubs but is surrounded by old forests. In spring, we observed woodland-associated species such as the Great tit and the Chaffinch breeding in the surrounding forests rather than at our study site, while Red-backed shrikes were breeding on the study site. The insectivorous shrikes usually prefer grazed open areas scattered with shrubs that provide nest sites and hunting perches (*Ceresa et al., 2012*) as well as insects associated with grazers (*Brambilla, Rubolini & Guidali, 2007*; *Vanhinsbergh, 1999*), which may explain the positive association of the woodland guild with grazers also during the breeding period. On the other hand, the higher number of woodland-associated birds counted on the study site during autumn and winter reflects that the resident Chaffinches and Great tits, which are predominantly granivorous in winter (*Perkins et al., 2000*), often searched for food on the grazed open areas. *Leal et al. (2019)* reported a similar pattern, and also concluded that woodland birds profit from the activity of grazers opening up the vegetation and thus increasing prey and seed visibility. Similar to our results, *Laiolo (2005)* found that open areas had greater avian diversity in winter than in spring, hosting woodland birds as well as other guilds.

Unlike birds associated with open areas and woodlands, the numbers of aerial- and wetland-birds seemed to be relatively independent of the variability of grazer densities at our study site. The guild of aerial-foraging birds did not show a clear association with

either grazing or land cover. As suggested by *Neilly & Schwarzkopf (2019)*, a reason for this may be that conspicuous responses of bird species to grazing are only to be expected in guilds that forage close to the ground. On the other hand, *Willi, Korner-Nievergelt & Grüebler (2011)* found that the presence of grazers is likely advantageous for swallows and swifts, due to the high abundance of flying insects in the proximity of large herbivores and *Musitelli et al. (2016)* similarly proposed a positive effect of high insect abundance around livestock farms on swallows. The overall weak relationship we found between the numbers of aerial-foraging birds and habitat variables may indicate that the distribution of flying insects was relatively even across land-cover types in our study site. Consistent with our findings, *Neilly & Schwarzkopf (2019)* found that the "abundance of aerial foraging guilds was not influenced by the grazing treatments, vegetation type or habitat variables". Further research could address the relationship between grazer densities and flying insects, taking differences in land cover into account. However, such an approach may need to account for variables that might be hard to control for, for instance, the effect of wind and temperature on both the habitat use of grazers and the distribution of insects.

Unsurprisingly, we found that in wetland birds, it was the presence of water rather than grazer density that explained their abundance over the grazed area. Both in the breeding period and in the migration period, we counted more birds of the wetland guild at low grazer densities (Figs. 1 and 2), but this may also indicate that, at our study site, horses and cattle spent only a small fraction of their time in or around water bodies (L.L. own observation). Also in the UK, Highland cattle in semi-wild conditions spent less than 0.5% of their daily time standing in water, while this behaviour was not at all observed in Konik horses (*Laidlaw, 2018*). In our study, it seemed that the primary factor influencing wetland birds was the presence of water within the grid cell rather than the presence of grazers.

The positive relationship between meadow cover and the number of bird observations in three of four guilds (the except being woodland birds) and the generally positive relationship between shrub cover and the number of individuals for all four guilds are in line with earlier studies. For example, *Pons et al. (2003)* pointed out that grasslands scattered with shrubs are of highest value for bird conservation, since these areas host more individuals and more species than homogenous forested areas.

Overall species richness showed an optimum at grid cells with low to moderate grazer densities, similar to findings from earlier studies which reported that many bird species favor areas with low-intensity grazing (*Barzan, Bellis & Dardanelli, 2021*). We found the highest species richness in our study site occurred where scattered woody patches and water bodies enriched the open habitats. Since such heterogeneous habitats are structurally complex and rich in resources (*Stein, Gerstner & Kreft, 2014*), they can host more individuals and support a greater species diversity (*Srivastava & Lawton, 1998*).

## CONCLUSIONS

Our results indicate that low-intensity mixed-species grazing fosters both the presence of individuals of birds and their overall species diversity, particularly benefiting the open-area foraging bird guild. Since aerial-, wetland- and woodland birds were also present in

high numbers in the grazed area and showed either positive or neutral relationships with grazer density, our findings indicate that low-intensity grazing fulfils the needs of birds with different foraging and breeding ecologies. We therefore conclude that the management of alluvial grasslands with low-intensity cattle and horse grazing resulting in heterogenous environments is a favourable conservation plan for ecosystem restoration and maintenance.

## ACKNOWLEDGEMENTS

We thank the team of the Réserve Naturelle Petite Camargue Alsacienne for making it possible to conduct our research in the nature reserve. We thank the reviewers Ask Herrik and Katja Irob for their constructive suggestions to improve the manuscript. Special thanks for Lily Fogg for the helpful comments and proofreading our manuscript.

### Funding

This work was supported by the Fondation de bienfaisance Jeanne Lovioz, the Foundation Emilia Guggenheim-Schnurr, the Ornithologische Gesellschaft Basel, the Swiss Association Pro Petite Camargue Alsacienne, the Foundation Wolfermann-Nägeli, the Foundation Frey-Clavel, and the MAVA Foundation. The funders had no role in study design, data collection and analysis, decision to publish, or preparation of the manuscript.

### Grant Disclosures

The following grant information was disclosed by the authors:
Fondation de bienfaisance Jeanne Lovioz.
Foundation Emilia Guggenheim-Schnurr.
Ornithologische Gesellschaft Basel.
The Swiss Association Pro Petite Camargue Alsacienne.
The Foundation Wolfermann-Nägeli.
The Foundation Frey-Clavel.
The MAVA Foundation.

### Competing Interests

Fränzi Korner-Nievergelt is employed by oikostat GmbH.

### Author Contributions

- Lilla Lovász conceived and designed the experiments, performed the experiments, analyzed the data, prepared figures and/or tables, authored or reviewed drafts of the article, and approved the final draft.
- Fränzi Korner-Nievergelt conceived and designed the experiments, analyzed the data, prepared figures and/or tables, authored or reviewed drafts of the article, and approved the final draft.
- Valentin Amrhein conceived and designed the experiments, authored or reviewed drafts of the article, and approved the final draft.

### Field Study Permissions

The following information was supplied relating to field study approvals (i.e., approving body and any reference numbers):

Field work was approved by the Réserve Naturelle National Petite Camargue Alsacienne

### Data Availability

The data and code is available at OSF: Lovász, Lilla, Fränzi Korner-Nievergelt, and Valentin Amrhein. 2023. "Effects of Grazer Density, Season and Habitat on Bird Guilds in a Restored Conservation Area." OSF. December 29. doi: 10.17605/OSF.IO/XJEBY.

### Supplemental Information

Supplemental information for this article can be found online at http://dx.doi.org/10.7717/peerj.17777#supplemental-information.

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
