# Peer review of "Natural grazing by horses and cattle promotes bird diversity in a restored European alluvial grassland"

_PeerJ, doi:10.7717/peerj.17777_

## Round 0.1 · original submission · Major Revisions

Dear Authors,

The manuscript was reviewed by two independent reviewers. They both found the manuscript interesting, but they also raised several points that need to be addressed by authors. Please, note that Reviewer#1 requested the text to be proof-reading by a fluent English speaker. The same reviewer suggested that 'results and discussion' section should be divided into two separate sections. Reviewer#2 pointed several parts of the text that need to improve clarity. I hope you find reviewers' comments useful.

**Language Note:** The Academic Editor has identified that the English language must be improved. PeerJ can provide language editing services - please contact us at [email protected] for pricing (be sure to provide your manuscript number and title). Alternatively, you should make your own arrangements to improve the language quality and provide details in your response letter. – PeerJ Staff

·

Basic reporting

The text would benefit greatly from proof-reading by a fluent English speaker or by use of a
language editing program such as Writefull. Just to get rid of small grammatical mistakes
throughout the text.

The 'results and discussion' section should be divided into two separate sections.

Experimental design

The study design, using gps to track the movements of the grazers, is a clever set-up, which is effective in precisely measuring the effects of grazer density on different bird guilds. However, you should really see if you can add grid ID as a random effect in your analysis or at least show that there is no spatial autocorrelation in the data to make the results more robust.

Validity of the findings

The study examines how large grazers affect bird species richness in a European grassland. This is
an interesting and increasingly relevant topic when it comes to rewilding European landscapes. However, the main hypothesis of the study, that “the presence of birds depends not only on grazer densities, but also on the season and on the characteristic of the habitat”, reveals that your focus may be a bit off the mark. As I see it, the most interesting finding of the study is the effect of grazer density on the different bird guilds as well as overall species richness, not so much the effect of season and land cover. Therefore, the results and discussion should focus on the grazing aspect and shorten the discussion of the other variables. The finding that low-density grazing increases bird species richness in alluvial ecosystems has important application potential for management of these ecosystems and should thus be given more attention in the discussion.

Additional comments

1: You could consider a title that summarize the main finding of the study in one sentence. Such as
“Low-intensity grazing facilitates open-area foraging birds in a restored European grassland” or
“Low-density grazing supports bird diversity in a restored European grassland”. This would be
more eye-catching and give a better indication of the findings.
25: Mention where the study took place.
27: *influence
29: Mention the study period and the size of the dataset – e.g., number of transects and bird
counts in total.
32: Explain what you used the GAMMs to model.
35: Use another word than ‘profit’. Maybe “select areas with low-intensity year-round grazing”.
36-37: Use ‘in’ instead of ‘on’: ‘in open landscapes’, ‘in the grazed area’ etc.
39: The conclusion could be more focused on management of grasslands: “Low-intensity grazing is
an important management tool in creating heterogeneity in grasslands, thereby facilitating habitat
and forage for a diverse bird species assemblage.”
46-48: “Were often shaped”. What period are you referring to here?
48: “Were high-conservation value”. Again, when?
49: Maybe use “hosting a diverse assemblage of species” instead.
52: *10-year study.
53: *bird species.
69: *Seem.
73: Which birds were found to benefit from the activities of grazers?
77-78: Use present tense in the introduction. You can use past tense when describing your own
results.
80: What is meant by groups? Maybe just use species or guilds.
83: Do grazers facilitate nesting for wetland birds, or do they simply minimize predation risk by
providing short vegetation with better view? I think it might be mixed a bit together here.
87: Between birds and *grazers.
100: You could use “In this study, we investigate how…” instead.
101: You could use ‘spatial distribution’ instead of ‘area use’.
100-112: A bit much methodology. Instead, you could simply state the objectives and hypotheses
of the study and save the details for the methods section.
110: What are your hypotheses?
112: Remember to stick to present tense throughout the introduction.
123: Oak is missing scientific name.
129: Remove “so-called”.
182: *For decades.
185: ‘GPS positions’ may be more intuitive than ‘fixes’.
219: How did you avoid auditory double counts?
245: *The fieldwork was carried out with the permission…
267: Guess this also depends on the season – i.e., the recovery time for the vegetation. However, I
think 3 weeks is a reasonable cut-off.
287: Remove ‘so-called’.
296: *another GAMM.
320: This section should be divided into two separate sections; first present the results with
descriptive statistics, then interpret and discuss them.
348: No need to discuss non-significant relationships.
351: However, your study did not find this relationship?
354: ‘Slightly more’ is too vague.
396: Is there strong correlation between grazing density and water?
458: Start the conclusion section by restating the hypotheses and summarizing the most important
finding. As I see it, the fact that low-density grazing increases bird species richness in alluvial
ecosystems is the most important finding of the study.
459: ‘Seemed to profit’ is also a bit vague. Your results clearly show that open-area foraging birds
select grazed areas.
465: I’m not surprised that the presence of birds depend on both season and habitat, however, I
think it’s much more interesting to focus on the effect of grazing on the different bird guilds
throughout the year and just keep season and habitat as additional explanatory variables.
491-493: The potential application of these findings for better managing alluvial ecosystems is an
important point that should be further discussed.
494-496: This is quite obvious and maybe not the best way to end the article. Instead, end with
the main conclusion of your study and its potential application by managers.
Figure 1: Very illustrative and a good way to present the results. However, it’s counter-intuitive
that the high number birds is lowest on the figure legend. This should of course be turned around.
Figure 3: Way too much information for one figure. Should be moved to supplementary materials.
Figure 5: I think this is an interesting result that should get more attention in the text. Here, you
show that, throughout the year, the greatest diversity of bird species was found in areas with lowdensity grazing. This finding should really be discussed more in detail.

·

Basic reporting

The study developed a comprehensive global model of terrestrial herbivore populations, considering eco-physiological traits of various species. The model, coupled with a global vegetation model, predicts the maximum potential and current biomass of herbivores, revealing significant declines in large herbivores and dominance of small species.
I generally believe this approach is interesting and could be used in the future to address important ecological topics.
However, several parts could be reduced and slightly restructured to for better clarity. Every section could be framed with a short overview of what is to come so that the reader does not get lost. The introduction should prepare the reader better about the potential relationship between grazer presents/impact on landscape and occurrence of birds. It would be interesting to get an overview of the different bird guilds to understand their behaviour and occurrence better.


Abstract:
19-20: Should the focus of the study be impacts of cattle or more on birds? You might have to rethink your opening sentence.
20-21: the connection to birds is a little bit weak. Try to relate it more to habitat requirements of birds, or why birds are impact by habitat changes (food/shelter/eggs)
24-28: These are not objectives but what you did. Objectives would be:
1. Determine the relationship between the abundance and species richness of birds across four foraging guilds and the area utilization patterns of Highland cattle and Konik horses within an alluvial grassland.
2. Examine the impact of seasonality and land cover on the spatial distribution of individuals and species within various bird guilds present in the grazed area.
30. Not clear to me how the hourly position of grazers relates to weekly transect counts of birds to answer and fulfill your objectives. Maybe more like this: We will utilize GPS-collars on all grazers to evaluate the frequency and density of their hourly positions, thereby assessing their area utilization patterns (grazing intensity?). Weekly transect counts of birds will be conducted to characterize their distribution within the grazed area, while land cover surveys will be undertaken to describe the habitat (to relate the bird occurrence to habitat type, maybe make clear what habitat type means?).
36: What do you mean bird species and individuals?
34-36: You relate this mostly to grazing intensity. It’s not clear how the occurrence relates to habitat type in terms of cover? Maybe clarify in the methods section better.
40: The wording heterogeneous landscape is very sudden. If you make a statement like this, you should also relate the results to it.
41: Same here with regards to feeding ecologies? You did not mention that you look at birds with different feeding habits. This is confusing.
I feel the abstract needs a bit more attention. It has to be rewritten in a way to prepare the reader better for what is to come.



Introduction:

45: The first sentence should describe all the players of the study and the problematic. You only say that grazing is a common management tool. Would rephrase to make it more interesting
49: you should talk especially about the diversity of birds species in grasslands here/give examples
52: you could prepare the reader here that you are next talking about birds. Maybe just by saying “....associated species, such as birds.
60-62: this sentence feels a little bit out of place, not exactly sure what you mean
64: what are landscape features? How do they affect species?
67: or the feeding behaviour? You mean “and” the feeding behaviour?
90: What exactly did they say about why grazed areas in both seasons are important for skylarks?
91: birds that are in enclosures? Or grazers in Enclosures?
100: you could improve the transition by starting this sentence with: In this study, we aim to address this gap by investigating how the species richness and abundance of individuals within a bird community….
105-108: Could be rephrased, maybe something along the lines like this: We focused on elucidating the relationships between the numbers of individuals and species within these four guilds and the density of cattle and horse GPS positions within our study area—a recently restored alluvial grassland within a French nature reserve. Additionally, we evaluated the dynamic interplay between bird and grazer densities across different seasons and varying land cover conditions.
110: the hypothesis is unclear, maybe: We hypothesize that the presence of birds within the grazed area is contingent upon not only grazer densities but also on seasonal variations and habitat characteristics, owing to the distinct ecological behaviors of birds across different foraging guilds.

Experimental design

The setup seems sound but the questions and gaps are not well framed yet.

134: name the source of the satellite imagery
135: The observer? Someone estimated the cover it in the field? Or you did it in QGIS with the help of digital tools? How often did you analyse the land cover characteristics?
I think an overview of when you did what and in which frequency at the beginning could be useful in general.
141: assumed or you determined?
145: I’m a bit confused with the observer walking along the grid cell. In the field? How did you know where the grid cell boundaries are?
146-148: suggest to move to beginning of paragraph
How do the land cover types relate to open/heterogeneity of habitat?
161-163: rephrase, e.g. To counteract the natural succession facilitated by willow and poplar saplings encroaching upon the meadows and to foster a heterogeneous environment, large grazers were progressively introduced to the test enclosure. The introduction began with horses in September 2018, followed by cattle in January 2019.
164: were increased sounds weird, as if some external force was responsible for the birth. Would rephrase
172: does this mean the initial cattle were only tracked for 6 months? Meaning you have data points of different lengths?
172-175: too long. Shorten
176: the two additional collars were used on cattle that were already collared or on additional cattle?
182-184: would mention this earlier
192: I would start the paragraph with someting like this sentence.. E.g. “To determine the density distribution of the animals over the study area, we recorded hourly GPS points…. “
204: how can birds show connection to the area?
213-216: It could be nice for the reader to have general information as what was when sampled and how often in the same place of the paragraph (e.g. always top)
219: cut-off distance?
226: is this the right wording? Foraging ecology means where they feed? And not what they feed?
233: “on a control non-grazed area”
241-242: Similar comment as before. Only once?
249: UTM grid?

Validity of the findings

Novelty missing although they could make a point.
Underlying data to my knowledge not provided. I did not see a data availability statement.
General links to research questions and hypothesis not well given.

The results/discussion part should state more clearly where their assumptions are based on, they used complicated statistics so they should be able to back up their results with statistical output. Very weak/hesitant wording (e.g. it seemed it increased) should be replaced with better language and statistics. Simultaneously, the link to other studies is very weak and rarely shown. Also the ecological significance is missing in many places. The results/discussion part seems like a mere iteration of results without going deep into the discussion part. The discussion itself was unfortunately not very well structured, the ecological significance and link to existing literature poorly provided. I made some suggestions but it should only serve as an orientation that should be applied to the rest of the manuscript. This whole section should be revised.
The conclusion is not a real conclusion but entails many parts that should be moved in the discussion part. Again, very weak and hesitant arguments that should be replaced by more concrete language. The whole conclusion should be rewritten. I stopped at one point, my earlier comments in the conclusion can be applied to the rest of the conclusion. The conclusion should finish with a broader application of their work and not an “This indicates …” sentence.

Results/Discussion
Relation to grazer density? Season? Presence of birds between one guild and another

Reading the results made me realise that you did not sufficiently enough talk about potential relationships of grazer density and birds in the introduction (e.g. more insects near ungulates, meadow height influencing foraging, etc)

331: if you combine results and discussion, I would expect further explanation of the patterns observed in this first paragraph
351: Should be elaborated further
360-363: would expect more explanation here is as well
379: it seemed it decreased? Or the statistical analysis backed it up. Generally you
383: You mention uncertainty a few times. What do you mean exactly? Variation? Too many confounding factors
406: is this a common phenomena? Have other studies observed this?
439: does it make sense to calculate season per grid cell? And not by site?

Conclusion
459: seemed to profit or profited?
461: seemed to be important or was important
462-465: should not be in the conclusion but discussion
465-467: rather discussion
468: seemed
470: may suggest

Additional comments

There are too many figures in the results, I believe not more than 4 or 5 figures should be shown. Information presented here can definitely be summarised into less figures.

Figures
2a:b: what do you mean by the segment of bird abundance?
So you only had the grazer densities 5 and 125? That’s a huge difference.
Y axis label: would change to number of bird individuals, could be confusing otherwise

3: A lot of information in one figure. Very hard to read. Could be changed into several figures or some of it moved into appendix.
Sub-titles naming the bird guild could be helpful too.

Fig. 4: Not really sure what this figure is meant to tell us. There is no difference. I suggest moving this to the appendix and making 2 figures of Fig. 3

Fig. 5: Number of bird species

---

## Round 0.2 · Minor Revisions

The revised manuscript was reviewed by the two original reviewers and both of them agreed that the manuscript has improved significantly. However, they also provided some additional suggestions to increase clarity for readers. I hope you find reviewers's comments useful.

·

Basic reporting

The authors have thoroughly revised the manuscript and addressed all of my previous comments. The text has been proof-read and now contains clear and professional English. I believe the manuscript has improved significantly and I only have a few additional comments, mainly relating to the article figures. Other than that, I believe the manuscript now meets the PeerJ criteria and is ready for publication.

Experimental design

The study design, using gps to track the movements of the grazers, effectively measures the effects of grazer density on different bird guilds in a European alluvial grassland.

Validity of the findings

The study examines how large grazers affect bird species richness in a European grassland. This is an interesting and increasingly relevant topic when it comes to rewilding European landscapes. The finding that low-density grazing increases bird species richness in alluvial ecosystems has important application potential for management of these ecosystems.

Additional comments

The term "natural grazing" should be explained more explicitly. Alternatively, you could use the term "low-intensity grazing".
Line 97: A study on* the impact...
Line 135-137: The hypothesis that the presence of birds will vary is a bit vague. Maybe just remove this sentence.
Line 147: Why not include a map of the study area? It would be good to include a map of the study area showing the transect locations and land cover types.
Line 203: across* the study area.
Figure 4: The legend should say number of bird species. The same goes for figure 5.

·

Basic reporting

The text and overall structure has improved significantly. There are still some minor grammatical flaws or too long sentences but the overall language is good. The separation of results and discussion has improved clarity and the flow of reading. I am also happy about the rewritten conclusion.
I only have minor comments, mostly language editing.

Experimental design

L360: Split:
The comprehensive land-cover survey was conducted once during the study period in summer 2020. Over the 1.5-year duration of the study, the observed large-scale habitat characteristics remained largely stable, as confirmed by weekly visual inspections.
412: Personal curiosity, did you observe a difference in the effect of different grazer types? Could be interesting for a future study ;-)

438-446: The handling and changing of collars still sounds a bit confusing to me and I’m wondering how much of this information is needed in the main text. The data collected did not differ depending on collar type I assume?

483: replace recommendations with “approach”
485: this sentence can be deleted, just add this citation to the previous sentence.

486-488: rephrase, e.g:
Our objective was to conduct surveys on one morning every two weeks from January 31, 2019, to July 24, 2019, and weekly from March 20, 2020, to March 22, 2021. Data collection occurred exclusively under favorable weather conditions, characterized by the absence of precipitation and wind speeds below 5m/s.

540: what was the main habitat in the control site?

545: I don’t think the information in the parentheses is needed.

568: If you have research permission certificate numbers for handling the collars and using the land it would be good to include here as well.

Validity of the findings

The most notable finding of this research concerns the influence of grazer density on various bird guilds and overall species richness. Particularly noteworthy is the revelation that low-density grazing enhances bird species richness in alluvial ecosystems, highlighting its significant implications for ecosystem management. This aligns nicely with previous studies finding positive effects of low grazer density. I believe the mixed grazer part is particularly interesting here and could be further investigated in future studies.

Additional comments

The abstract is a lot nicer and informative now, but it is very long (>380 words). Some information can be combined. For example the objective and methods section. Something like “To examine the relationship between … we used GPS collars on all grazers to… “ as an example.

Introduction:

L137: Could be rephrased, eg:
However, ongoing human impact during the Anthropocene (Lewis & Maslin 2015), such as river regulations and agricultural intensification, persists in driving the decline of alluvial grasslands and the species dependent on them.

L141-145: Separate into two sentences.
Therefore, the European LIFE program (Silva 2008), focuses on the restoration of alluvial grasslands. The integration of grazing by ungulates, including horses and cattle, emerges as a significant management strategy, particularly in light of the expanding scope of rewilding initiatives (Linnartz & Meissner, 2014; Naundrup & Svenning, 2015; Pereira & Navarro, 2015).

L148: In this context, heterogeneity refers to the structural complexity of the land cover, leading to richness in resources.

L182: How are the grazers' activity linked to more food availability?

L308. make a new sentence after “area,”.
Additionally, we expected grazing to exert the most significant influence on bird density in open habitats.

Discussion:
1122: Split.
A similar pattern was observed in a riparian environment concerning livestock grazing in an experimental context (Nelson et al., 2011). However, to our understanding, our study marks the first attempt to compare nature conservation areas with and without mixed-species conservation grazing.

L1268: In our study, it seemed that the primary factor influencing wetland birds was the presence of water within the grid cell rather than the presence of grazers.

L1275: what is the bird conservation index? Are of highest value for bird conservation?

Conclusion:
1295: Our results indicate that low-intensity mixed-species grazing fosters both the presence of individual bird species and their overall diversity, particularly benefiting the open-area foraging bird guild.

External reviews were received for this submission. These reviews were used by the Editor when they made their decision, and can be downloaded below.

---

## Round 0.3 · accepted · Accept

Authors addressed all reviewers' comments and the revised manuscript is improved compared with the original version.

External reviews were received for this submission. These reviews were used by the Editor when they made their decision, and can be downloaded below.